# Propagation of a Fatigue Crack Through a Hole

**DOI:** 10.3390/ma17246261

**Published:** 2024-12-21

**Authors:** Diogo Neto, Joel Jesus, Ricardo Branco, Edmundo Sérgio, Fernando Antunes

**Affiliations:** University Coimbra, Centre for Mechanical Engineering, Materials and Processes (CEMMPRE), Department of Mechanical Engineering, 3030-788 Coimbra, Portugal; diogo.neto@dem.uc.pt (D.N.); joel.jesus@uc.pt (J.J.); ricardo.branco@dem.uc.pt (R.B.); edmundo.sergio@uc.pt (E.S.)

**Keywords:** fatigue, stop-hole, initiation life, crack propagation rate, cumulative plastic strain

## Abstract

The stop-hole technique is a well-known strategy to extend the fatigue life of cracked components. The ability to estimate fatigue life after the hole is important for safety reasons. The objective here is to develop strategies for the accurate prediction of initiation and propagation life ahead of the stop-hole. Experimental work was developed in a Compact-Tension (CT) specimen made of 7050-T7451 aluminium alloy and with a 3 mm diameter hole. A total number of 625,000 load cycles were required to re-initiate the crack after the hole. Crack initiation life after the hole was estimated using the Theory of Critical Distances combined with the Smith–Watson–Topper parameter. A value of a_0_ = 31.83 µm was obtained for El Haddad parameter, which was used to define the critical distance. The predicted life was found to be only 4% lower than the experimental value. The fatigue crack growth (FCG) rate was calculated using a node release strategy, assuming that cyclic plastic deformation is the main damage mechanism and that cumulative plastic strain is the crack driving parameter. A good agreement was found between the numerical predictions of da/dN and the experimental results. The main result, however, is the proposed methodology, which allows predicting the initiation and propagation lives in notched components.

## 1. Introduction

Different strategies have been used to increase fatigue life in cracked components, namely, hole drilling, crack filling [1,2,3,4], composite patches [5,6,7], welding repair [8,9], overloading [10,11], Brinell indentation at corner points [12,13,14,15,16], cold expansion [17,18,19], laser shock peening [20,21,22], shot peening [23,24,25], or spot heating [26,27,28].

The crack tip hole is a classical strategy to stop crack propagation, used by many maintenance crews all over the world since it is relatively inexpensive, simple, and fast to apply [29,30]. This technique is widely employed for the cracks which are detected in aircraft skin and also in steel bridges [31,32,33]. In fact, the drilling of a hole, destroying the crack front, replaces a sharp crack by a round-tip notch. The propagation ahead of the hole requires the re-initiation of the crack. Makabe et al. [34] proposed the arrest of crack growth not only drilling a stop hole but also inserting a pin. Two alternative hole drilling strategies are deflecting holes and crack flank holes. The deflecting holes are used to change the crack growth direction and also arrest the fatigue crack growth (FCG). Drilling two crack flank holes symmetrically relative to the crack line is the other method to reduce the stress intensity factor, to decrease the crack growth rate and to increase the fatigue life of the cracked part [35,36,37]. Song et al. [29] studied how stop-crack hole can improve the crack initiation life and fatigue crack growth (FCG) life of aluminium alloy and stainless-steel structures. They found that the larger the diameter of the hole, the longer the life. Fu et al. [38] studied the stop-crack effect of holes with different positions and diameters. It was found that the best position of the hole is behind the crack, with the hole edge coinciding with the crack tip. Hu et al. [39] studied the effects of hole location, shape, and size on residual fatigue life. The results showed that the larger the diameter of the stop hole, the greater the range of influence on the residual fatigue life. Shin et al. [35] suggested that the use of hole drilling for extending the fatigue life is more effective than infiltration and safer than applying overload. However, the hole drilling technique can only be used for cracks with nearly straight fronts, therefore is not applicable to corner or surface cracks.

It is important to be able to predict fatigue life after the use of hole-drilling technique. This knowledge is also important for cracks crossing holes in riveted or bolted joints. The determination of stress–life curves (usually named S-N curves) is probably the most popular method for fatigue analysis of structural components. However, for having accurate results, a separated and detailed analysis of initiation and propagation is recommended. Propagation is assumed to be dominant in components with defects or sharp notches, while initiation is expected to be dominant in smooth components or having low concentration factor notches. The main objective here is to propose a strategy to predict fatigue life in a stop-hole, involving the components of initiation and propagation. Experimental work was developed in a CT specimen made of 7050-T7451 aluminium alloy, with a 3 mm diameter hole. The experimental test was replicated numerically.

## 2. Experimental Work

### 2.1. Experimental Procedure

Figure 1 shows the geometry of the CT specimen, having a width W = 36 mm and a thickness of 2 mm. A hole with a diameter of 3 mm was placed ahead of the initial notch so that the crack can propagate through it. This hole was made with a rotary drill, and the distance between the extremity of the initial notch and the centre of the hole was 6 mm. A similar specimen without stop-hole was also tested.

The material used was the 7050-T7451 aluminium alloy. Table 1 and Table 2 show the chemical composition and the mechanical properties of this material, respectively. The ratio between the tensile strength and the yield stress is 1.1. This material is expected to have a behaviour typical of metallic materials; therefore, it can be considered representative.

The experimental tests were made according to ASTM E647 [41] in an electrical testing machine (INSTRON ELECTROPLUS model E10000, Norwood, Massachusetts, EUA), with a load capacity of 10 kN. Table 3 presents the load parameters considered before and after the hole. The fatigue test was initiated with ∆K_0_ = 6 MPa·√m, but after the hole the load range was increased to ∆K_0_ = 12.49 MPa·√m to promote the re-initiation of the crack. A sinusoidal load wave was applied with a loading frequency of 15 Hz. The crack length was measured with a travelling microscope (Carl Zeiss AG®, Oberkochen, Baden-Württemberg, Germany) with a magnification of 45× and a resolution of 10 µm. The number of load cycles was registered at increments of 200 µm, in order to obtain the crack length, *a*, versus number of load cycles, *N*, curve. Finally, the FCG rate was obtained using the 5-point polynomial incremental technique.

### 2.2. Experimental Results

Figure 2 presents the crack length, *a*, versus the number of load cycles, *N*. Before the hole, the crack length increased non-linearly, as it normally does. In the first point registered in Figure 2, the crack has already propagated 1 mm from the notch. The hole occupies the crack length range between 11.5 and 14.5 mm, as is indicated. There are no experimental measurements in the range a = 9.2–11.5 mm because the propagation was very fast in the region just before the hole.

The hole stops the crack propagation, and a significant number of load cycles is required to re-start the propagation. In fact, 570,000 load cycles were applied with maximum and minimum loads of 650 and 32.5 N, respectively, as indicated in Table 3. However, no signs of initiation were detected. Therefore, in order to initiate the crack, the load was increased by 25%, and more 55,000 load cycles were applied to see crack propagation again. After the hole, the propagation was faster, and the non-linearity of a-N curve was also observed but was less evident.

Figure 3 presents FCG rate obtained from a-N plots in the CT specimens with and without stop-hole. Before the hole the da/dN-ΔK curve is linear in log-log scales. These results are coincident with those obtained in a specimen without hole, as could be expected. After the hole da/dN is higher than before the hole, as could be expected since the crack length is higher. However, the values are below those without hole. This seems to indicate a smoothing influence of the notch over the stress singularity. Note that the K solution available for the CT specimen was always used, which does not include the presence of the hole. This may explain the difference between da/dN results ahead of the hole obtained without and without hole.

## 3. Prediction of Re-Initiation Life

### 3.1. Methodology

The crack initiation life after the hole was estimated using the Theory of Critical Distances combined with the Smith–Watson–Topper parameter. In this study, the Line Method was used, which states that failure occurs when a critical length in the notch-controlled process zone undergoes a critical value of the *SWT* parameter. This methodology, schematically illustrated in Figure 4, has been successfully introduced in a recent paper by the authors [42].

The *SWT* parameter represents the product of both the maximum stress (*σ_max_*) and the strain amplitude (*ε_a_*) of the loading cycle, and can be defined as follows:(1)$$SWT=\sigma_{max\;}\varepsilon_{a},$$

The critical value of the Smith–Watson–Topper (*SWT*) parameter, termed here as *SWT*_eff_, was obtained using the following equation:(2)$${SWT}_{eff}=\frac{1}{2a_{0}}\int_{0}^{{2a}_{0}}SWT\left(r\right)dr,$$
where a_0_ is the material characteristic length [43], and *SWT*(r) is the distribution of the *SWT* parameter along a straight line emanating from the notch bisector up to a distance 2a_0_. Regarding the material characteristic length (a_0_), the El-Haddad equation was used:(3)$$a_{0}=\frac{1}{\pi }{\left(\frac{{\Delta K}_{th}}{{\Delta \sigma }_{0}}\right)}^{2}$$
where ΔK_th_ represents the stress intensity factor range threshold, and Δσ_0_ is the material plain fatigue limit. The *SWT* parameter can also be determined from cyclic material properties as follows:(4)$$SWT=\frac{{\sigma_{f}^{\prime }}^{2}}{E}{\left({2N}_{f}\right)}^{2b}+\sigma_{f}^{\prime \;}\varepsilon_{f}^{\prime }{\left({2N}_{f}\right)}^{b+c}$$

### 3.2. Finite Element Analysis

The experimental work developed in CT specimen with stop hole was replicated numerically in terms of geometry, loading, and material properties. The finite element code DD3IMP, originally developed to simulate deep drawing [44], was adapted for the simulation of FCG due to the recognised competence in the simulation of plastic deformation.

A plane stress state was assumed, considering the relatively small thickness of the CT specimens tested experimentally. Accordingly, the CT specimen was simulated with a thickness of only 0.1 mm. Since the experimental work was developed with 2 mm CT specimens, the loads presented in Table 3 were divided by 20.

The proper modelling of elastic-plastic behaviour of the AA7050-T7451 is crucial for the accuracy of predictions. The elastic behaviour was defined by the Hooke’s law, using the Young modulus (E) and Poisson ratio (ν), both listed in Table 2. The plastic behaviour was described with the von Mises yield criterion coupled with a mixed hardening model using the Swift isotropic and Armstrong–Frederick kinematic hardening laws, under an associated flow rule. The Swift isotropic hardening law [45] is given by:(5)$$Y=K{\left({\left(\frac{Y_{0}}{K}\right)}^{\frac{1}{n}}+{\bar{\varepsilon }}^{p}\right)}^{n},$$
where *Y* is the flow stress and ${\bar{\varepsilon }}^{p}$ denotes the equivalent plastic strain. The material parameters of Swift law are *K*, *Y*_0_ and *n*. The Armstrong–Frederick law describes the non-linear kinematic hardening as follows [46]:(6)$$\dot{X}=C_{X}\left[X_{\mathrm{Sat}}\frac{\sigma^{\prime }-X}{\bar{\sigma }}-X\right]{\dot{\bar{\varepsilon }}}^{p},$$
where X is the back-stress tensor, $\bar{\sigma }$ is the equivalent stress, $\sigma^{\prime }$ is the deviatoric Cauchy stress tensor, ${\dot{\bar{\varepsilon }}}^{p}$ is the equivalent plastic strain rate and $C_{X}$ and $X_{\mathrm{Sat}}$ are material parameters. Table 4 presents the different material parameters, which were fitted using low-cycle fatigue experimental results.

Figure 5 presents the finite element mesh of the CT specimens used in numerical analysis, namely the specimen without hole (Figure 5a) and the specimen with hole (Figure 5b,c). Linear hexahedral finite elements were used to describe the CT specimens, considering reduced selective integration. The mesh was refined in the crack propagation region and enlarged in remote regions to reduce the total number of nodes and elements. Moreover, in the case of the specimen with stop hole, two different meshes were generated in order to locate the mesh refinement before (Figure 5b) and after (Figure 5c) the hole. The smallest elements had a size of 8 × 8 µm^2^. Only one element was considered along the thickness of the specimen. Since crack propagation was simulated by node release, each crack increment had an extent of 8 µm, i.e., the size of crack front elements.

### 3.3. Calculation of Initiation Life

Table 5 presents the cyclic properties of the 7050-T7451 aluminium alloy, determined by Hou et al. [48] from low-cycle fatigue tests (R_ε_ = −1), considering strain amplitudes in the range 0.3–1.0%. The analysis was carried out using the half-life cycle associated with each strain level which was assumed to be representative of the stabilised behaviour. Figure 6 shows the evolution of the *SWT* parameter with the number of reversals to failure for the tested aluminium alloy based on the low-cycle fatigue tests.

The material characteristic length (El-Haddad parameter, a_0_) was calculated assuming that ΔK_th_ = 2.50 MPa m^0.5^ and Δσ_0_ = 250 MPa, respectively [49,50], and a_0_ = 31.83 µm.

The distributions of the *SWT* parameter along the hole bisector, i.e., a *SWT*(r) functions, for the two loading blocks are presented in Figure 7. As can be seen, the maximum value of the *SWT* parameter occurs at the hole surface (d = 0) and then decreases smoothly as the distance increases. In this case, the effective value of the *SWT* parameter for the first loading block (*SWT*_eff,1_), computed from the *SWT*(r) function using Equation (1), was equal to 0.781 MPa. This calculation is graphically illustrated in Figure 7 by the grey lines. Regarding the second block, as expected, the values of the *SWT*(r) function for the same value of d/a_0_ were higher, which is explained by the greater loading level of this loading block. This also results in a higher effective value of the *SWT* parameter (*SWT*_eff,2_ = 0.888 MPa).

The crack initiation life associated with a specific value of *SWT*_eff_ can be estimated through Equation (2). In this research, as described above, the crack initiation resulted from the application of a low-high loading sequence. The first block was applied during 570 × 10^3^ cycles without any evidence of macroscopic crack, followed by a second block applied during 56 × 10^3^ cycles, from which a crack with a length of 0.284 mm was observed. In order to estimate the damage caused by each loading block, the Miner–Palmgren rule [51] was used:(7)$$\frac{n_{1}}{N_{f,1}}+\frac{n_{2}}{N_{f,2}}=1,$$
where n_1_ and n_2_ are the number of cycles applied at the 1st and 2nd loading levels, and *N_f_*_,1_ and *N_f_*_,2_ are the crack initiation lives at the 1st and 2nd loading levels, respectively. Therefore, the crack initiation life associated with a crack length equal to a_0_ is estimated by summing both n_1_ and n_2_.

Based on Figure 6, it can be concluded that the damage accumulation resulting from the first loading block (*SWT*_eff,1_ = 0.781 MPa) was significant. At this loading level, the number of cycles to crack initiation (*N_f_*_,1_) is 640.80 × 10^3^ cycles (i.e., 2*N_f_*_,1_ = 1.282 × 10^6^ reversals). This value is relatively close to the number of cycles applied in the first block (n_1_ = 570 × 10^3^ cycles) resulting in a fatigue damage of 88.95%. Regarding the second loading block, if no damage is assumed, *N_f_*_,2_ = 305.87 × 10^3^ cycles (i.e., 2*N_f_*_,2_ = 611.75 × 10^6^ reversals to failure). As expected, as the effective value of the *SWT* parameter increased (*SWT*_eff,2_ = 0.888 MPa) relative to the first block, the number of cycles to crack initiation (*N_f_*_,2_) decreased. Solving Equation (4) for n_2_, considering a fatigue damage of 11.05% caused by the second block, results in 33.79 × 10^3^ cycles, corresponding to a crack initiation life equal to 603.795 × 10^3^ cycles, which is quite close to the experimental value (626.42 × 10^3^ cycles). The experimental value is approximately 4% higher than the predicted value, but, as mentioned above, it is associated with a longer crack length (a_i_ = 0.284 mm).

Fatigue crack initiation is a complex phenomenon affected by diverse factors, such as material inhomogeneity, residual stresses, defects and imperfections, loading variations, surface finishing, among others. Therefore, the error obtained here is quite acceptable. Ling et al. [52] focused on the fatigue crack initiation life of 7075 aluminium alloy, a closely related material. They reported that the “predicted fatigue life reaches 69.1–87.3% of the experimental measurement” which is less accurate than the present study.

## 4. Numerical Prediction of FCG

A numerical approach was followed to calculate the propagation life. A node release strategy was considered, assuming that cyclic plastic deformation is the main damage mechanisms and that cumulative plastic strain is the crack driving parameter. As the specimen is cyclically loaded, the plastic deformation accumulates up to a critical value, which is supposed to be a material property. The FCG rate is the ratio between the crack increment (8 µm) and the number of load cycles needed to accumulate the critical value. The critical value of cumulative plastic strain was calculated in a previous work of the authors, being 80% for the AA7050-T7541 [53]. This approach was validated in previous works, comparing the numerical predictions of da/dN with experimental results obtained for load blocks [54] and overloads [55].

Figure 8 presents the numerical predictions of da/dN, which are compared with the experimental results. Before the hole, there is a good agreement between the numerical and experimental results. However, the numerical da/dN has a fast (asymptotic) increase in the region immediately before the hole. The experimental results do not show this trend, mainly because there are no measurement points in this region. After the hole, the FCG rate is relatively small which may be explained by the smoothing effect of the hole over the crack tip stress field. As the crack propagates ahead of the hole, there is a fast increase in da/dN approaching the experimental trend.

The numerical model may be affected by the assumption that FCG is only linked with cyclic plastic deformation; however, there is a general agreement about the dominance of this damage mechanism in Paris law-regime. The size of crack tip elements is also a major parameter. The FCG rate is obtained using:(8)$$\frac{da}{dN}\approx \frac{∆a}{∆N}$$
where Δa is the size of crack tip elements and ΔN is the number of load cycles required to reach the critical cumulative plastic strain. The reduction in element size, and therefore of Δa, increases the crack tip plastic deformation reducing the number of load cycles required to reach the critical strain value. Therefore, the model is quite robust and almost insensitive to element size. Finally, the modelling of elastic-plastic behaviour is always questionable. The identification of material parameters was made using low-cycle results, which is the procedure usually followed. On the other hand, the experimental work is also affected by problems, namely by the relatively number of measurement points. After the stop-hole, the measurement of crack length only started at 350 μm. Therefore, the trends obtained numerically before and after the stop-hole are expected to be more realistic than the trends obtained experimentally.

## 5. Conclusions

A strategy is proposed here to calculate the fatigue life after a stop-hole. This approach calculates the initiation and propagation lives separately.

Experimental work was initially developed in a CT specimen made of 7050-T7451 aluminium alloy with a 3 mm hole. A total number of 625,000 load cycles were required to re-initiate the crack after the hole. 

The estimation of crack initiation life after the hole was made using the Theory of Critical Distances combined with the Smith–Watson–Topper parameter. A value of a_0_ = 31.83 µm was obtained for El Haddad parameter, which was used to define the critical distance. The predicted value of initiation life was found to be 4% lower than the experimental value.

The fatigue crack growth rate was calculated using a node release strategy, assuming that cyclic plastic deformation is the main damage mechanism and that cumulative plastic strain is the crack driving parameter. A good agreement was also found with experimental results.

## Figures and Tables

**Figure 1 materials-17-06261-f001:**
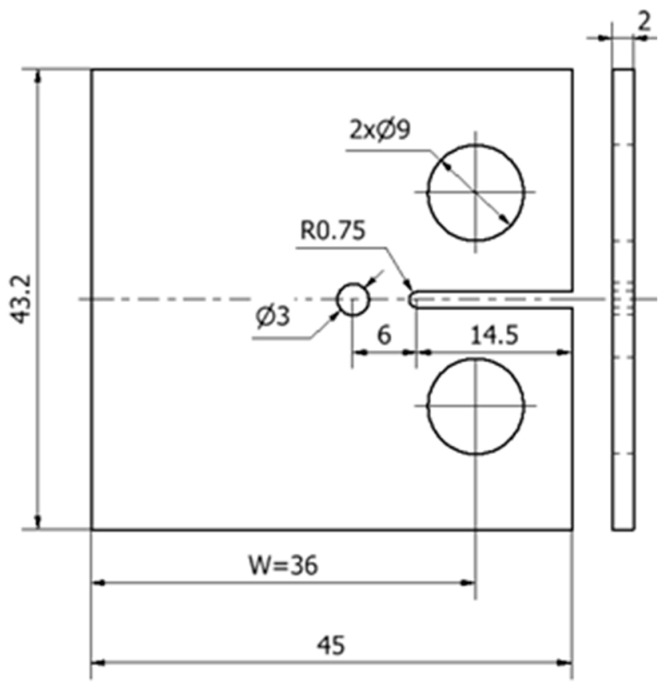
Geometry of the C(T) specimen with stop-hole (dimensions in mm).

**Figure 2 materials-17-06261-f002:**
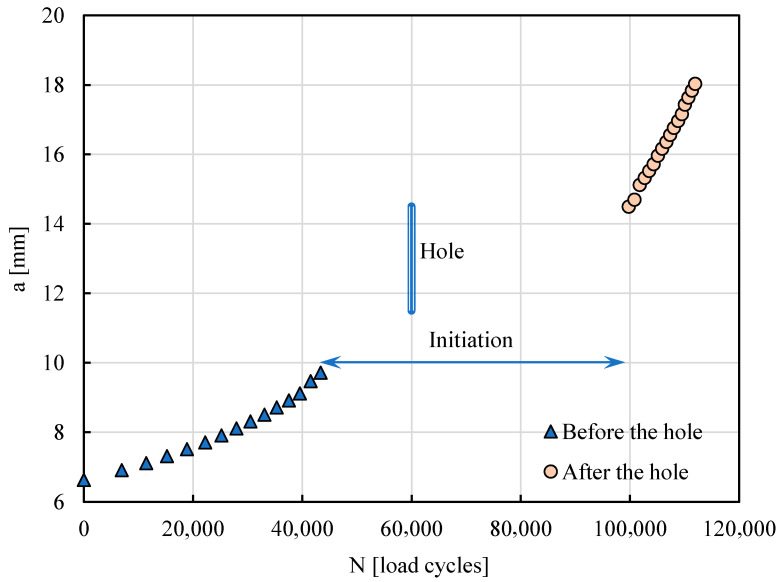
Crack length versus number of load cycles.

**Figure 3 materials-17-06261-f003:**
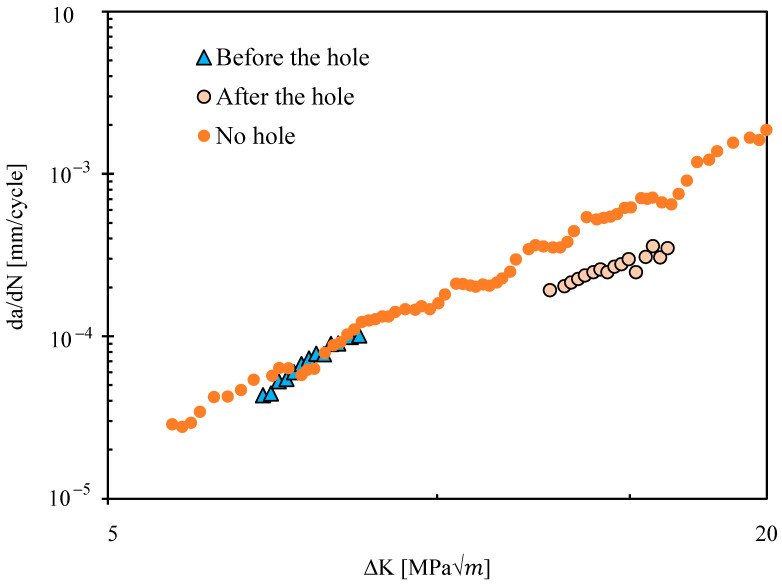
Fatigue crack growth rate (da/dN) versus ΔK.

**Figure 4 materials-17-06261-f004:**
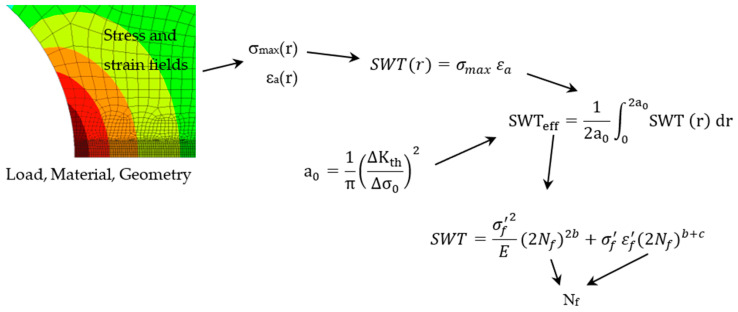
Strategy for the prediction of initiation life (*N_f_*).

**Figure 5 materials-17-06261-f005:**
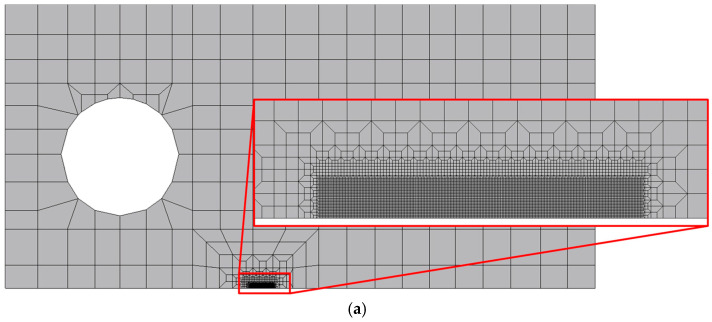
Finite element mesh of the CT: (**a**) specimen without hole; (**b**) specimen with hole (mesh refinement before the hole); (**c**) specimen with hole (mesh refinement after the hole).

**Figure 6 materials-17-06261-f006:**
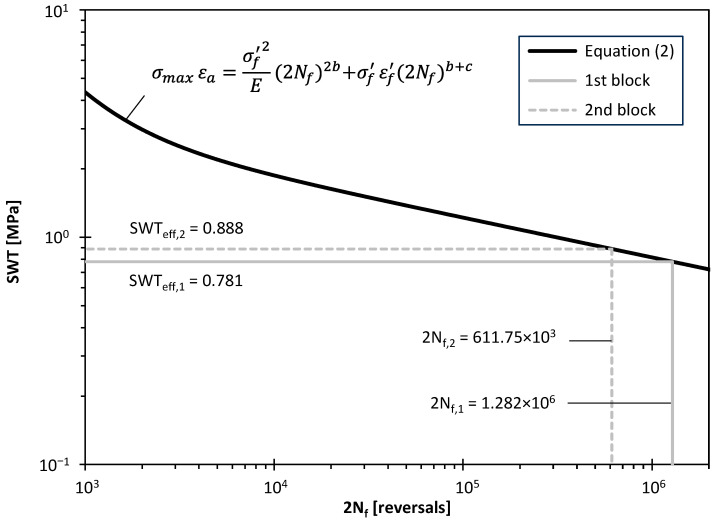
*SWT* parameter versus number of reversals to failure derived from low-cycle fatigue tests performed by Hou et al. [47].

**Figure 7 materials-17-06261-f007:**
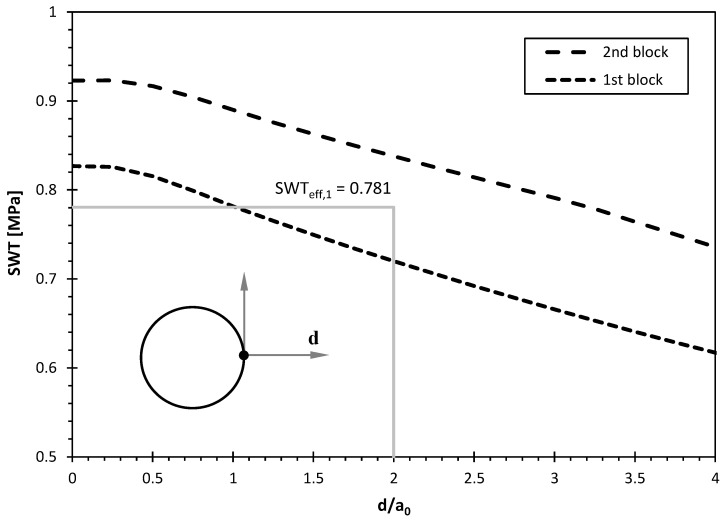
Distribution of the *SWT* parameter along a straight line emanating from the hole bisector for both loading blocks applied.

**Figure 8 materials-17-06261-f008:**
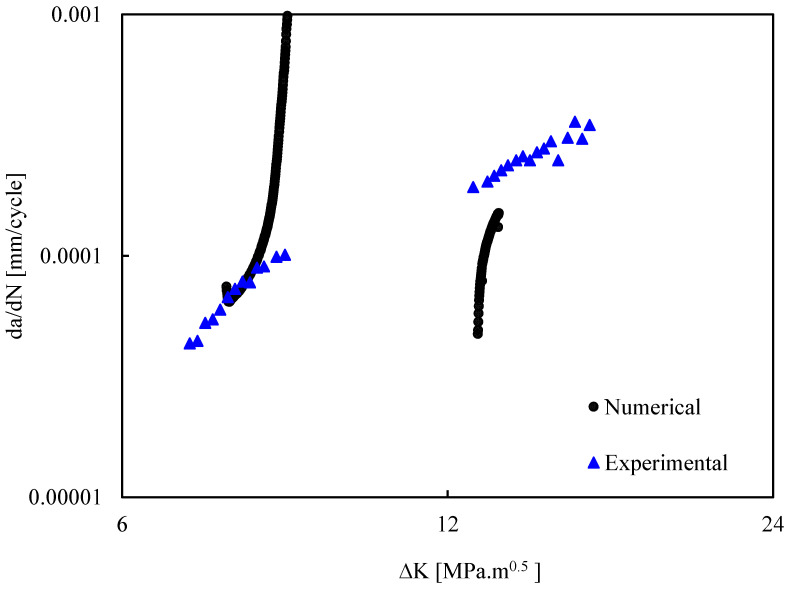
Numerical predictions versus experimental results of da/dN for the CT specimen with hole.

**Table 1 materials-17-06261-t001:** Composition of the AA7050-T7451 (weight %) [40].

Al	Cu	Cr	Mg	Mn	Ti	Si	Fe	Zn
Bal.	2.17	<0.01	2	0.01	0.04	0.04	0.06	6.67

**Table 2 materials-17-06261-t002:** Mechanical properties of the AA7050-T7451 [40].

Density	Young Modulus	Yield Stress	Tensile Strength	Poisson Ratio
2.75 g/cm^3^	71.7 GPa	469 MPa	524 MPa	0.33

**Table 3 materials-17-06261-t003:** Loading parameters.

	Before the Hole	After the Hole
F_max_	650 N	770 N
F_min_	32.5 N	38.5 N
ΔK	6 MPa·√m	12.49 MPa·√m
R	0.05	0.05

**Table 4 materials-17-06261-t004:** Parameters of the constitutive model of the AA7050-T7451 [47].

Material	*Y* _0_	*C_x_*	*X* _sat_
L-T orientation	357.93	378.64	183.24

**Table 5 materials-17-06261-t005:** Fatigue-strength and fatigue-ductility constants for the AA7050-T7451 aluminium alloy.

Young Modulus	*σ′_f_*	*b*	*ε′_f_*	*c*
72 GPa	808.86 MPa	−0.0873	70.858	−1.4294

## Data Availability

Data available on request from the authors.

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
