# Peer review of "Propagation of a Fatigue Crack Through a Hole"

_materials, 2024, doi:10.3390/ma17246261_

Round 1
Reviewer 1 Report
Comments and Suggestions for Authors
The paper deals with a study on the propagation of a fatigue crack through a hole. The authors aimed to develop strategies for accurate prediction of initiation and propagation life ahead of the stop-hole. A good agreement was found between numerical predictions and experimental results. The research has some significance, and some results were obtained. However, some problems need to be modified before final acceptance.
1. The important value and significance of this study can be emphasized in the abstract.
2. Whether the selected test material is representative and whether it has inspiration for other materials?
3. The predicted value was found to be 4% lower than the experimental value. What are the main causes of errors, and are there methods and strategies to further reduce them?
Reviewer 2 Report
Comments and Suggestions for Authors
The paper deals with the analysis of the effectiveness of the stop-hole technique in extending the fatigue life of structural elements. The authors propose a methodology combining the critical distance theory and the Smith-Watson-Topper parameter to predict the fatigue life after the application of a stop hole, verifying the results using experiments and numerical simulations. The research includes the initiation and propagation of a crack in samples made of AA7050-T7451 aluminium, subjected to cyclic loads.
The adopted goal of the work can be considered new and significant, because it combines elements of previously used techniques in a way that allows for more precise prediction of fatigue durability. The authors effectively justify their approach, and the presented results indicate the innovative nature of the study.
Experimental validation (Fig. 8) showed significant discrepancies between the authors' research results and experimental data, both in quantitative and qualitative terms. In the experiment, the relationship between crack propagation rate (da/dN) and ΔK is linear, while the numerical results show asymptotic behaviour with a local discontinuity in the area of the stopping hole. These discrepancies indicate the need to verify the model assumptions. They may be the adopted material reinforcement models or interpretations of the critical distance. The authors should additionally use other models, then compare the results and choose the best one. The work also requires supplementing information on the number of experimental samples, a more detailed description of numerical assumptions, and an analysis of the influence of mesh density on simulation results.
Reviewer 3 Report
Comments and Suggestions for Authors
Abstract
Line 11, It is suggested to define an acronym the first time it appears in the article, such a CT.
Line 50 Consider changing the word “Anyway” to start this paragraph, it seems that this word reduces the importance of what was said.
Line 52 Consider explaining the meaning of the expression or acronym S-N since it is the first time it is mentioned.
Line 53 Is the expression better accuracy redundant? Is accuracy already a noun that does not need to be qualified by an adjective? Will your idea be properly transmitted if you eliminate “better”?
Line 59-60 Consider changing the word “situation” for a less trivial word, for example “phenomenon”
Line 63 It seems confusing the explanation in this line about the width of the CT specimen at 36mm and the dimensions of 43,20 and 45,00 should one of those be the width? Also instead of commas, should you use a decimal period for scientific notation? The units should be noted in the figure as well or ultimately clarify in the title below what are the units.
Line 69 and Tables 1 and 2 What is the source of this information? Add reference.
Line 78 Add city of manufacture for the Instron Electroplus machine.
Line 82 and 83 Indicate make, model and city of manufacture of the travelling microscope.
Line 83 check the meaning of the word selected: application, should it be amplification?
Line 85-86 check syntax of the sentence that starts after the period (Finally…)
Line 90 check grammar or syntax: “AS is usual” seems to lack a noun.
Lines 102 to 109
This section talks about a comparison of experiments or results of a specimen without hole, add reference of that other study, when or by who was it made, it this was part of this work, then that is not clearly explained. Is all of this a description of a “situation” or what would be a better word for a scientific paper?
Figure 2 is best if located after the text where it is explained and before the text that explains Figure 3.
Since it is apparent that Figure 3 is describing behaviours in two different specimens (also see the comments to lines 102 to 109), then each specimen shall be properly identified.
Line 124-125 review English use.
Lines 283 to 288
This paragraph is from the template, if no supporting data is provided then this paragraph left as it was in the template does note make sense.
Delete it or provide the supporting data location.
Round 2
Reviewer 2 Report
Comments and Suggestions for Authors
The authors have carefully considered the suggestions and criticisms and modified the manuscript accordingly. It is the opinion of the present reviewer that the paper should now be accepted for publication as is.
Reviewer 3 Report
Comments and Suggestions for Authors
The authors have made the requested improvem